# Frequency and mortality rate following antimicrobial-resistant bloodstream infections in tertiary-care hospitals compared with secondary-care hospitals

Cherry Lim[1,2]*, Viriya Hantrakun[1], Preeyarach Klaytong[1], Chalida Rangsiwutisak[1], Ratanaporn Tangwangvivat[3], Chadaporn Phiancharoen[3], Pawinee Doung-ngern[3], Somkid Kripattanapong[3], Soawapak Hinjoy[3], Thitipong Yingyong[3], Archawin Rojanawiwat[4], Aekkawat Unahalekhaka[4], Watcharaporn Kamjumphol[4], Kulsumpun Khobanan[4], Pimrata Leethongdee[4], Narisorn Lorchirachoonkul[5], Suwimon Khusuwan[6], Suwatthiya Siriboon[7], Parinya Chamnan[8], Amornrat Vijitleela[9,10], Traithep Fongthong[11], Krittiya Noiprapai[11], Phairam Boonyarit[11], Voranadda Srisuphan[11], Benn Sartorius[2,12,13], John Stelling[14], Paul Turner[2,15], Nicholas P. J. Day[1,2], Direk Limmathurotsakul[1,2,16]*

1 Mahidol-Oxford Tropical Medicine Research Unit, Faculty of Tropical Medicine, Mahidol University, Bangkok, Thailand, 2 Centre for Tropical Medicine and Global Health, University of Oxford, Oxford, United Kingdom, 3 Department of Disease Control, Ministry of Public Health, Nonthaburi, Thailand, 4 Department of Medical Sciences, Ministry of Public Health, Nonthaburi, Thailand, 5 Department of Medicine, Ratchaburi Hospital, Ratchaburi, Thailand, 6 Department of Medicine, Chiangrai Prachanukroh Hospital, Chiang Rai, Thailand, 7 Department of Medicine, Sunpasitthiprasong Hospital, Ubon Ratchathani, Thailand, 8 Department of Social Medicine, Sunpasitthiprasong Hospital, Ubon Ratchathani, Thailand, 9 Department of Medical Services, Ministry of Public Health, Nonthaburi, Thailand, 10 National Health Security Office, Nakhonsawan, Thailand, 11 The Office of Permanent Secretary, Ministry of Public Health, Nonthaburi, Thailand, 12 Centre for Clinical Research (UQCCR), School of Medicine, University of Queensland, Brisbane, Australia, 13 Department of Health Metric Sciences, Faculty of Medicine, University of Washington, Seattle, WA, United States of America, 14 Brigham and Women's Hospital and Harvard Medical School, Boston, MA, United States of America, 15 Cambodia-Oxford Medical Research Unit, Angkor Hospital for Children, Siem Reap, Cambodia, 16 Department of Tropical Hygiene, Faculty of Tropical Medicine, Mahidol University, Bangkok, Thailand

* direk@tropmedres.ac (DL); cherry.lim@ndm.ox.ac.uk (CL)

**Data Availability Statement:** All relevant data are within the manuscript and its Supporting

## Abstract

There are few studies comparing proportion, frequency, mortality and mortality rate following antimicrobial-resistant (AMR) infections between tertiary-care hospitals (TCHs) and secondary-care hospitals (SCHs) in low and middle-income countries (LMICs) to inform intervention strategies. The aim of this study is to demonstrate the utility of an offline tool to generate AMR reports and data for a secondary data analysis. We conducted a secondary-data analysis on a retrospective, multicentre data of hospitalised patients in Thailand. Routinely collected microbiology and hospital admission data of 2012 to 2015, from 15 TCHs and 34 SCHs were analysed using the AMASS v2.0 (www.amass.website). We then compared the burden of AMR bloodstream infections (BSI) between those TCHs and SCHs. Of 19,665 patients with AMR BSI caused by pathogens under evaluation, 10,858 (55.2%) and 8,807 (44.8%) were classified as community-origin and hospital-origin BSI, respectively. The burden of AMR BSI was considerably different between TCHs and SCHs, particularly of

Information files. The study used openly anonymous AMR surveillance reports generated from each hospital which are available at: https://doi.org/10.6084/m9.figshare.20318193.

**Funding:** The study was supported by the DDC, MoPH, Thailand, and Defense Threat Reduction Agency (DTRA), U.S.. This research was funded in part by the Wellcome Trust (224681/Z/21/Z and Wellcome Trust Institutional Translational Partnership Award-MORU). CL is supported by the Wellcome Trust (106680/B/14/Z). BS is supported by a grant from the UK Department of Health and Social Care using UK aid funding managed by the Fleming Fund (R52354 CN001). For the purpose of Open Access, the author has applied a CC BY public copyright licence to any Author Accepted Manuscript version arising from this submission. The funders had no role in study design, data collection and analysis, decision to publish, or preparation of the manuscript.

**Competing interests:** The authors have declared that no competing interests exist.

hospital-origin AMR BSI. The frequencies of hospital-origin AMR BSI per 100,000 patient-days at risk in TCHs were about twice that in SCHs for most pathogens under evaluation (for carbapenem-resistant *Acinetobacter baumannii* [CRAB]: 18.6 vs. 7.0, incidence rate ratio 2.77; 95%CI 1.72–4.43, p<0.001; for carbapenem-resistant *Pseudomonas aeruginosa* [CRPA]: 3.8 vs. 2.0, p = 0.0073; third-generation cephalosporin resistant *Escherichia coli* [3GCREC]: 12.1 vs. 7.0, p<0.001; third-generation cephalosporin resistant *Klebsiella pneumoniae* [3GCRKP]: 12.2 vs. 5.4, p<0.001; carbapenem-resistant *K. pneumoniae* [CRKP]: 1.6 vs. 0.7, p = 0.045; and methicillin-resistant *Staphylococcus aureus* [MRSA]: 5.1 vs. 2.5, p = 0.0091). All-cause in-hospital mortality (%) following hospital-origin AMR BSI was not significantly different between TCHs and SCHs (all p>0.20). Due to the higher frequencies, all-cause in-hospital mortality rates following hospital-origin AMR BSI per 100,000 patient-days at risk were considerably higher in TCHs for most pathogens (for CRAB: 10.2 vs. 3.6, mortality rate ratio 2.77; 95%CI 1.71 to 4.48, p<0.001; CRPA: 1.6 vs. 0.8; p = 0.020; 3GCREC: 4.0 vs. 2.4, p = 0.009; 3GCRKP, 4.0 vs. 1.8, p<0.001; CRKP: 0.8 vs. 0.3, p = 0.042; and MRSA: 2.3 vs. 1.1, p = 0.023). In conclusion, the burden of AMR infections in some LMICs might differ by hospital type and size. In those countries, activities and resources for antimicrobial stewardship and infection control programs might need to be tailored based on hospital setting. The frequency and in-hospital mortality rate of hospital-origin AMR BSI are important indicators and should be routinely measured to monitor the burden of AMR in every hospital with microbiology laboratories in LMICs.

## Introduction

Understanding and monitoring the burden of antimicrobial resistant (AMR) bacterial infection is important to design strategies for interventions [1]. A recent modelling study estimated that there are 1.27 million deaths attributable to AMR infections comparing to non-AMR infections in 2019 globally [2]. The study also highlighted the limited availability of data in LMICs, [2] where most of the currently available data were from university hospitals and tertiary-care hospitals (TCHs) [3, 4].

Multiple parameters are required for monitoring and evaluating the AMR burden in hospital settings. The proportions (%) of patients with growth of AMR strains of bacterial species (over total number of patients with growth of bacterial species; i.e. AMR proportion [7]) are commonly used to represent AMR burden [5, 6]. However, the AMR proportion alone cannot reflect the burden of AMR in absolute terms. For example, consider two hospitals with comparable activity, size and target population. Both hospitals may have the same proportion of bloodstream infections (BSI) due to methicillin-resistant *Staphylococcus aureus* (MRSA), which is 20%. However, the MRSA burden in hospital A would be considerably lower than that in hospital B, as hospital A has 10 MRSA cases out of 50 *S. aureus* BSI, whereas hospital B has 20 MRSA cases out of 100 *S. aureus* BSI. The frequencies of patients with AMR infections within a population during a reporting period (i.e. AMR frequencies [7]) are other important parameters. The AMR frequencies, typically expressing per 1000, 1,000 or 100,000, are also commonly used to monitor, evaluate and compare the AMR burden between hospitals or survey sites regardless of the size or services of the hospital [8–10].

AMR proportions and AMR frequencies are reported to be different by type and size of hospitals in some settings in high-income countries (HICs). In Spain, Oteo *et al.* reported that

the proportion of MRSA is higher in hospitals with >500 beds than in those with <500 beds [11]. In Germany, Said *et al.* reported that the proportion of carbapenem-resistant *Acinetobacter baumannii* (CRAB) is higher in TCHs and secondary-care hospitals (SCHs) compared to outpatient clinics [12]. In the U.S., Gandra *et al.* reported that the proportion of AMR infections is not different between TCHs and small community hospitals [13]. The point prevalence survey of healthcare-associated infections (HAIs) in European acute care hospitals shows that HAI prevalence is highest in hospitals with ≥650 beds and lowest in those with <200 beds, [14] suggesting that frequency of HAI (per patients who were admitted to the hospital) is associated with hospital size.

We recently developed the AutoMated tool for Antimicrobial resistance Surveillance System (AMASS), an offline application to generate standardized AMR surveillance reports from routinely available microbiology and hospital data files, and independently tested the application in seven hospitals in seven countries [15]. The automatically generated reports stratify infections into community-origin and hospital-origin based on the recommendations of World Health Organization Global Antimicrobial Resistance and Use Surveillance System (WHO GLASS), and provide additional metrics on mortality involving AMR and non-AMR BSI [15]. Collaborating with Ministry of Public Health (MoPH) Thailand, we previously obtained and analysed microbiology and hospital admission data files of 60 hospitals from 2012 to 2015 in Thailand, and reported the burden of melioidosis, an infectious disease caused by *Burkholderia pseudomallei*, in Thailand [16].

The aim of this study was to examine the burden of AMR BSI in TCHs and compare that with SCHs using AMASS [15] on the microbiology and hospital admission data from 2012 and 2015 in Thailand. We also examine the burden of culture-confirmed notifiable bacterial diseases under evaluation using the reports generated by the AMASS.

## Methods

### Study setting

In 2012, Thailand had a population of 64.4 million, consisted of 77 provinces, and covered 513,120 km$^2$. In each province, there is at least one SCH or TCH, [17] equipped with a microbiology laboratory capable of performing bacterial culture using standard methodologies for bacterial identification and susceptibility testing provided by the Bureau of Laboratory Quality and Standards, MoPH, Thailand [18]. The health systems in each province were integrated into 12 groups of provinces, known as health regions, plus Bangkok as health region 13. For example, SCHs generally referred patients to TCHs within the same health region. In 2012, there were 96 public hospitals (68 SCHs and 28 TCHs) in health regions 1 to 12 in Thailand.

### Study design

We conducted a retrospective, multicentre surveillance study of all SCHs and TCHs in health regions 1 to 12 in Thailand. From the hospitals that agreed to participate, data were collected from microbiology and hospital admission data files between January 2012 and December 2015 as previously described [16]. Variables in the microbiology data file included patient hospital number (HN), specimen type, specimen collection date, culture result, and antibiotic susceptibility testing result, and each row contained information for each specimen. Variables in the hospital admission data file included HN, admission date, discharge date, and in-hospital discharge outcome, and each row contained information for each admission. Each data set was analysed using the AMASS v2.0, [19] and HN was used as a record linkage between the two data files of each hospital. The AMASS analysed the data and generated reports based on the recommendation of WHO GLASS [20]. A deduplication process was automatically

conducted in which only the first isolate of a species per patient per specimen type per survey period was included in the report [15]. The AMASS v2.0 included an additional report on notifiable bacterial diseases (Annex A) and blood culture contamination rate (Annex B) (S1 Text in S1 File). The statistics in the AMR surveillance reports (in PDF and CSV format) were then extracted for analysis. The reports generated by the AMASS were validated using two methods: (a) checking data verification log files generated by the AMASS whether all information was imported accurately (e.g. total number of specimens, total number of hospital admissions, number of missing values, total number of isolates per organism in the raw microbiology data file, and total number of antibiotics being tested) [16] and (b) comparing the summary data and reported generated by the AMASS with data generated from manual calculations obtained from complete line listing of several organisms.

For AMR infections, we analysed the following organisms: CRAB and carbapenem-resistant *Pseudomonas aeruginosa* (CRPA), third-generation cephalosporin-resistant *Escherichia coli* (3GCREC) and *Klebsiella pneumoniae* (3GCRKP), carbapenem-resistant *E. coli* (CREC), carbapenem-resistant *K. pneumoniae* (CRKP) and MRSA which are in the WHO GLASS list of priority AMR bacteria [21] and are of local importance. Only blood culture isolates were included in the analysis.

## Definitions

AMR BSI is defined as a case of infection in patients with blood culture positive for CRAB, CRPA, 3GCREC, 3GCRKP, CREC, CRKP or MRSA. Non-AMR BSI is defined as cases of infection in patients who had blood culture positive for carbapenem-susceptible *A. baumannii* (CSAB), carbapenem-susceptible *P. aeruginosa* (CSPA), third-generation cephalosporin-susceptible *E. coli* (3GCSEC) or *K. pneumoniae* (3GCSKP), or methicillin-susceptible *S. aureus* (MSSA).

Community-origin BSI was defined for patients with first positive blood specimens in the hospital taken within the first two calendar days of admission with calendar day one equal to the day of admission [20]. Patients with first positive blood specimens taken after the first two calendar days were categorized as cases of hospital-origin BSI. The classification of community-origin and hospital-origin BSI was performed within AMASS and based on specimen dates and hospital admission dates extracted from the microbiology and hospital data files, respectively.

The proportion of AMR (%) was calculated as the percentage of patients with new AMR BSI over all patients with new BSIs for each pathogen of interest during the reporting period [19]. The frequency of AMR BSI for each pathogen of interest was calculated as the total number of new patients with AMR BSI during the reporting period per 100,000 admissions (for community-origin BSI), per 100,000 patient-days at risk (for hospital-origin BSI), and per 100,000 tested population (for community-origin and hospital-origin BSI). In-hospital mortality (%) following AMR BSI for each pathogen of interest was calculated as the percentage of patients with new AMR BSI who died in the hospitals. In-hospital mortality rates following AMR BSI for each pathogen of interest were calculated as the total number of patients with new AMR BSI who died in the hospitals during the admission following AMR BSI per 100,000 admissions (for community-origin BSI) and per 100,000 patient-days at risk (for hospital-origin BSI).

In the AMASS v2.0, blood culture contamination is defined as isolation of one or more common commensal organisms; including *Arcanobacterium* spp., *Arthrobacter* spp., *Bacillus* spp. (except *B. anthracis*), *Brevibacillus* spp., *Brevibacterium* spp., *Cellulomonas* spp., *Cellulosimicrobium* spp., *Corynebacterium* spp. (except *C. diphtheriae*, *C. jeikeium*, *C.*

*pseudotuberculosis*, *C. striatum*, *C. ulcerans*, and *C. urealyticum*), *Cutibacterium* spp., *Dermabacter* spp., *Dermacoccus* spp., *Diphtheroids* spp., *Exiguobacterium* spp., *Geobacillus* spp., *Helcobacillus* spp., *Janibacter* spp., *Knoellia* spp., *Kocuria* spp., *Kytococcus* spp., *Leifsonia* spp., *Microbacterium* spp., *Micrococcus* spp., *Nesterenkonia* spp., *Paenibacillus* spp., *Propionibacterium* spp., *Pseudoclavibacter* spp., *Staphylococcus* spp. (except *S. aureus* and *S lugdunensis*), *Trueperella* spp., *Virgibacillus* spp., and Viridans group *Streptococci* [18]. The blood culture contamination rate is defined as the ratio of the number of blood cultures with common commensal organisms over the total number of blood cultures.

The AMASS v2.0 also generated a summary report of culture-confirmed notifiable bacterial diseases caused by 11 pathogens including *Brucella* spp., *B. pseudomallei*, *Corynebacterium diphtheriae*, *Neisseria gonorrhoeae*, *Neisseria meningitidis*, Non-typhoidal *Salmonella* spp., *Salmonella* enterica serovar Paratyphi, *Salmonella* enterica serovar Typhi, *Shigella* spp., *Streptococcus suis*, and *Vibrio* spp.. The summary report included the total number and in-hospital mortality of patients with culture-confirmed notifiable bacterial diseases.

## Statistical analysis

We compared proportion, frequency, mortality and mortality rate of AMR BSI between SCHs and TCHs for each pathogen of interest. We preliminarily compared AMR proportions and mortality using Chi-square or Fisher's exact test when small samples (i.e. one or more expected values was <5 observations) and measurements from continuous variables using the Krustal-Wallis test. Then, we estimated the differences in proportions and mortality of AMR BSI between SCHs and TCHs using mixed-effect logistic regression models for patients nested within hospital using the STATA command xtlogit. We estimated the differences in frequency and mortality rate of AMR BSI between SCHs and TCHs using mixed-effect Poisson regression models for patients nested within hospital using the STATA command xtpoisson. We used mixed-effect models to estimate the fixed effects of hospital type while taking account for the random effects of patient within the same hospital. Multivariable mixed-effect models were also performed to control for other variables, including blood culture utilization rates of the hospitals and the health region where the hospitals located. These variables were included in the multivariable models because they could be associated with the hospital type and the outcome variables. We adjusted for blood culture utilization rate because if blood culture utilization rate was low, the observed AMR proportions could be higher than the true susceptibility profiles of pathogenic organisms [22]. Additionally, the observed AMR frequency per 100,000 admissions and per 100,000 patient-days could be lower than the true incidence rate of AMR infections [22]. We adjusted for health regions to control for potential variations in the AMR proportions and frequency across different regions. These regional differences could be influenced by other factors such as habits of antibiotic use and economic levels which were not explored in this study. There were no other potential confounders that were evaluated in the analysis. We then calculated total number and in-hospital mortality of patients with culture-confirmed notifiable bacterial diseases under evaluation. We used STATA (version 17.0; College Station, Texas) for the final statistical analyses and R version 4.0.5 for figures.

## Ethical considerations

Ethical permission for this study was obtained from the Institute for the Development of Human Research Protection, the Ethics Committee of the Faculty of Tropical Medicine, Mahidol University (MUTM 2014-017-01). Written approval was given by the directors of the hospitals to use their routine hospital admission database for research. Consent was not sought

from the individual patients as this was a retrospective study. This approach was approved by the Ethical and Scientific Review Committees.

## Results

### Baseline demographics

Ninety-five (99%) hospitals, out of 96 Thai hospitals that we approached, agreed to participate in the study. Twenty-five hospitals were excluded because either the microbiology or hospital admission data file was not available. Next, twenty-one hospitals were excluded because, in the microbiology data files, the antimicrobial susceptibility tests results were not available, readable or interpretable. Forty-nine hospitals were included in the analysis. A total of 35 hospitals (71%) had four years (from 2012 to 2015) of data available for analysis, four hospitals (8%) had three years, four hospitals (8%) had two years, and six hospitals (12%) had one year of data for analysis (S1 Table in S1 File).

Of 49 hospitals included in this study, 15 (31%) and 34 (69%) were TCHs and SCHs respectively (Fig 1). The median bed number was 672 (range 522–1,000) in TCHs and 335 (range 150–549) in SCHs (p<0.001). The median number of hospital admissions per year was 48,836 (range 30,409–98,428) in TCHs and 25,827 (range 7,221–62201) in SCHs (p<0.001). The total number of admissions was 3,134,815 in TCHs and 2,867,762 in SCHs. The blood culture utilization rate (median blood culture utilization rate 69 vs. 60 per 1,000 patient-days, respectively, p = 0.12) and the blood culture contamination rate (median blood culture contamination rates 4.1% and 3.6%, respectively, p = 0.94) were not statistically different between TCHs and SCHs. The all-cause in-hospital mortality among all patients admitted to TCHs was higher than those to SCHs (median in-hospital mortality 3.7% vs. 2.9%, respectively, p = 0.05).

### Proportion of AMR BSI

For community-origin BSI, there were differences in the proportions (%) of BSI being caused by AMR strains for some pathogens between TCHs and SCHs. The proportions of CRAB (36.9% vs. 25.0%, p = 0.041), 3GCREC (37.7% vs. 31.2%, p = 0.020) and 3GCRKP (24.4% vs. 19.2%, p = 0.053) were higher in TCHs than those in SCHs (Fig 2A and S2 Table in S1 File). The proportions of CRPA (18.4% vs. 17.1%), CREC (0.9% vs. 1.6%) and CRKP (2.3% vs. 2.7%) and MRSA (11.7% vs. 11.2%) were not significantly different between TCHs and SCHs.

For hospital-origin BSI, the proportions of AMR BSI for all pathogens, including CRAB (75.5% vs. 68.3%), CRPA (36.7% vs. 36.7%), 3GCREC (53.9% vs. 53.5%), 3GCRKP (63.7% vs. 57.3%), CREC (3.0% vs. 3.2%), CRKP (9.2% vs. 8.6%) and MRSA (37.4% vs. 28.0%), were not significantly different between TCHs and SCHs (Fig 3A and S2 Table in S1 File).

Similar findings were also observed in the multivariable models except that the proportion of CRAB for community-origin BSI was not significantly different between TCHs and SCHs (p = 0.66, S2 Table in S1 File).

### Frequency of AMR BSI

We next calculated the frequency of AMR BSI in SCHs and TCHs. For community-origin BSI, of all pathogens under evaluation, 3GCREC had the single highest frequency of AMR BSI per 100,000 admissions in both TCHs and SCHs (Fig 2B and S3 Table in S1 File). The frequencies per 100,000 admissions of community-origin BSI caused by CRAB (14.2 vs. 4.8, p<0.001), and 3GCRKP (30.5 vs. 18.8, p = 0.0017) in TCHs were relatively higher than those in SCHs. The frequencies of CRPA (7.1 vs. 4.9), 3GCREC (142.3 vs. 108.4), CREC (2.9 vs. 4.7), CRKP (2.4 vs 2.2) and MRSA (14.0 vs. 10.1) were not significantly different between TCHs and SCHs.

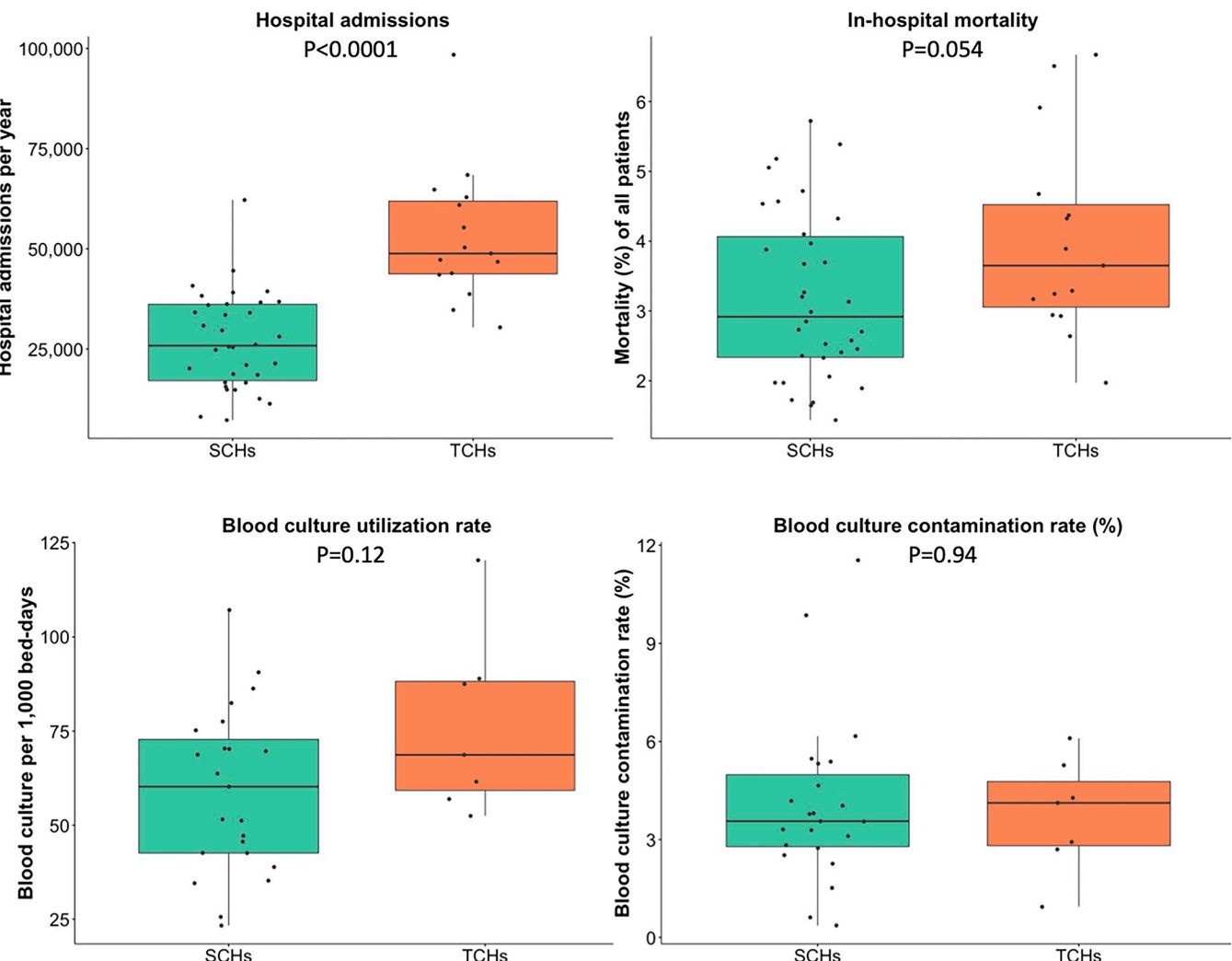

**Fig 1. Baseline demographics of 15 tertiary-care hospitals and 34 secondary-care hospitals in Thailand.** Each black dot represents the value reported by each hospital. Comparison was made using the Kruskal-Wallis test. The unit of analyses was a hospital. Blood culture (BC) utilisation rate (per 1,000 bed-days) and BC contamination rate (%) were estimated from 7 TCHs and 23 SCHs of which the microbiology data obtained included the culture results of "no growth".

Overall, the number of patient-days at risk of hospital-origin BSI included in the analysis was 12,341,585 in TCHs and 9,988,198 in SCHs. For hospital-origin BSIs, of all pathogens under evaluation, CRAB, 3GCREC and 3GCRKP were the top three organisms with the highest frequency in both TCHs and SCHs (Fig 3B and S4 Table in S1 File). Strikingly, the frequency of hospital-origin BSI per 100,000 patient-days at risk in TCHs was about twice that in SCHs for most of the pathogens under evaluation, including CRAB (18.6 vs. 7.0, incidence rate ratio [IRR] 2.77; 95%CI 1.72–4.43, p<0.001), CRPA (3.8 vs. 2.0, IRR 2.14; 95%CI 1.23–3.74, p = 0.0073), 3GCREC (12.1 vs. 7.0, IRR 1.80; 95%CI 1.29–2.50, p<0.001), 3GCRKP (12.2 vs. 5.4, IRR 2.23; 95%CI 1.57–3.17, p<0.001), CRKP (1.6 vs. 0.7, IRR 2.10; 95%CI 1.02–4.35, p = 0.045) and MRSA (5.1 vs. 2.5, IRR 1.88; 95%CI 1.17–3.01, p = 0.0091), except CREC (0.5 vs. 0.4, p = 0.39).

The frequencies of AMR BSI per 100,000 tested patients were calculated for 7 (47%) TCHs and 23 (68%) SCHs which provided results for blood cultures without growth (S5 Table in

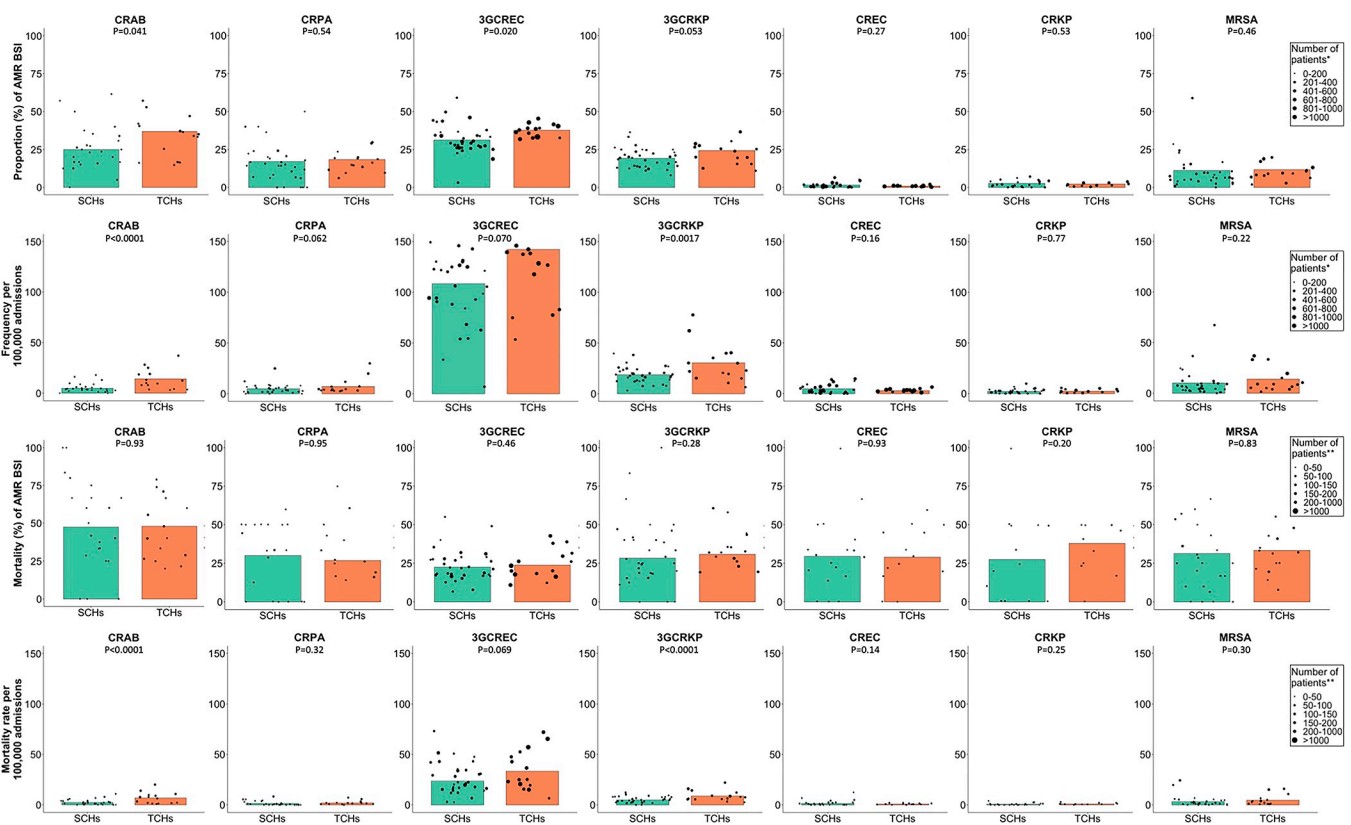

**Fig 2. Proportion (%), frequency (number of patients per 100,000 admissions), mortality (%) and mortality rate (number of deaths per 100,000 admission) of community-origin AMR BSI in 15 tertiary-care hospitals and 34 secondary-care hospitals in Thailand.** Each black dot represents the value reported by each hospital. The sizes of black dots are based on the total number of patients with blood culture positive for the bacterial species (row 1) and the total number of patients with blood culture positive for the AMR pathogens (row 2–4). Comparison was made using mixed-effect logistic or Poisson regression models of patients nested within hospitals.

S1 File). Similar findings of difference between TCHs and SCHs were also observed, but wider 95%CI and larger p values were observed.

Similar findings were also observed in the multivariable models except that the frequency of CRAP and CRKP for hospital-origin BSI per 100,000 patient-days at risk was not significantly different between TCHs and SCHs (p = 0.068 and p = 0.11, respectively) (S3-S5 Tables in S1 File).

## All-cause in-hospital mortality (%) following AMR BSI

Of 19,665 patients with AMR BSI, 10,858 (55.2%) and 8,807 (44.8%) were classified as community-origin and hospital-origin BSI, respectively (S6 Table in S1 File). Of 10,858 patients with community-origin AMR BSI, 2,873 (27.5%) died. Of 8,807 patients with hospital-origin AMR BSI, 3,874 (38.2%) died.

For both community-origin and hospital-origin BSIs, of all pathogens under evaluation, mortality (%) following AMR BSI caused by CRAB was higher than AMR BSI caused by other pathogens (Figs 2C and 3C). For both community-origin BSI and hospital-origin BSI, the differences in the mortality (%) following AMR BSI caused by all pathogens between TCHs and SCHs were not statistically significant.

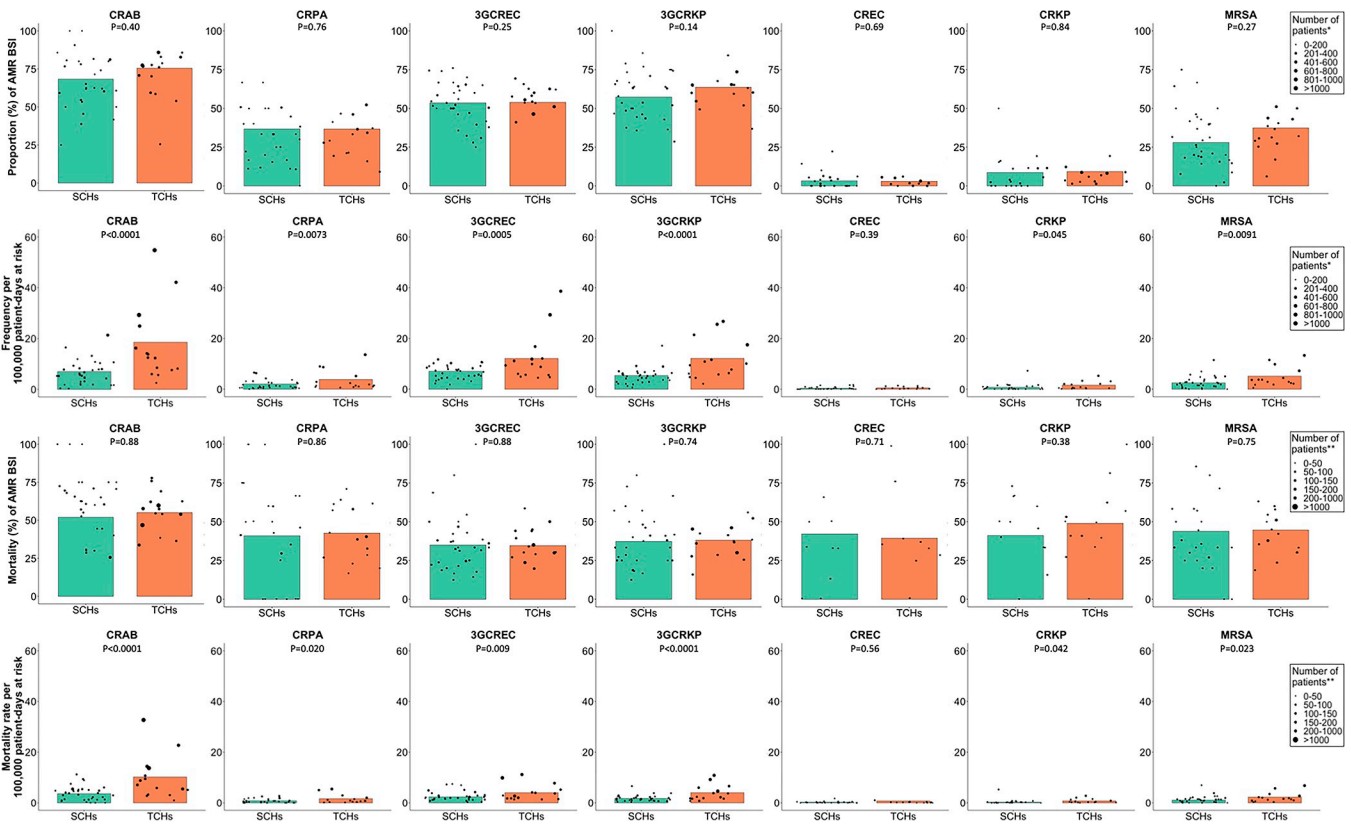

**Fig 3. Proportion (%), frequency (number of patients per 100,000 patient-days at risk), mortality (%) and mortality rate (number of deaths per 100,000 patient-days at risk) of hospital-origin AMR BSI in 15 tertiary-care hospitals and 34 secondary-care hospitals in Thailand.** Each black dot represents the value reported by each hospital. The sizes of the black dots are based on the total number of patients with blood culture positive for the bacterial species (row 1) and the total number of patients with blood culture positive for the AMR pathogens (row 2–4). Comparison was made using mixed-effect logistic or Poisson regression models of patients nested within hospitals.

Similar findings were also observed in the multivariable models except that the all-cause in-hospital mortality following AMR BSI caused by CRAB for hospital-origin BSI was higher in TCHs than that in SCHs (55.1% vs. 51.9%, p = 0.04, S6 Table in S1 File).

## All-cause in-hospital mortality rate following AMR BSI

We next calculated all-cause in-hospital mortality rate following AMR BSI in SCHs and TCHs. For community-origin BSI, of all pathogens under evaluation, 3GCREC had the single highest mortality rate following AMR BSI per 100,000 admissions in both TCHs and SCHs (Fig 2D and S7 Table in S1 File). The mortality rate following community-origin CRAB and 3GCRKP BSI per 100,000 admissions in TCHs was relatively higher than that in SCHs (6.8 vs. 2.3; mortality rate ratio [MRR] 2.92, 95%CI: 1.65–5.19, p<0.001 and 8.8 vs. 4.9; MRR 1.79, 95%CI 1.28–2.52, p<0.001, respectively). The differences in the mortality rate following community-origin CRPA, CREC, CRKP and MRSA BSI per 100,000 admissions between TCHs and SCHs were not statistically significant.

For hospital-origin BSI, of all pathogens under evaluation, CRAB had the single highest mortality rate per 100,000 patient-days at risk in both TCHs and SCHs (Fig 3D and S8 Table in S1 File). Strikingly, the mortality rate following hospital-origin AMR BSI caused by most of pathogens under evaluation per 100,000 patient-days at risk was also about two to nearly three times those in SCHs; including CRAB (10.2 vs. 3.6; MRR 2.77, 95%CI 1.71–4.48,

p<0.001), CRPA (1.6 vs. 0.8, p = 0.020), 3GCREC (4.0 vs. 2.4, p = 0.009), 3GCRKP (4.0 vs. 1.8, p<0.001), CRKP (0.8 vs. 0.3, p = 0.042) and MRSA (2.3 vs. 1.1, p = 0.023), though not for CREC (0.2 vs. 0.2, p = 0.56).

Similar findings were also observed in the multivariable models except that the mortality rate following community-origin 3GCREC BSI per 100,000 admissions in TCHs was relatively higher than in SCHs (33.4 vs. 23.7; adjusted MRR 1.21; 95%CI 1.03–1.41, p = 0.017) and the mortality rate following hospital-origin CRPA and CRKP BSI per 100,000 admissions in TCHs was not significantly different between TCHs and SCHs (adjusted MRR 1.97; 95%CI 0.83–4.63, p = 0.12, and adjusted MRR 1.81; 95%CI 0.80–3.63, p = 0.098, respectively).

## Notifiable bacterial diseases

Utilizing microbiology data and hospital admission data, we also calculated total number and in-hospital mortality of patients with culture-confirmed notifiable bacterial diseases indicated in the National Notifiable Disease Surveillance system (Report 506) of Thailand, [23] in the 49 hospitals from 2012 to 2015 (S9 Table in S1 File). The disease with the highest total number of cases was non-typhoidal *Salmonella* (NTS) infection (n = 11,264 patients), followed by melioi-dosis (an infection caused by *B. pseudomallei*; n = 6,164 patients) and *Vibrio* spp. infections (n = 2,143 patients).

The disease with the highest total number of all-cause in-hospital deaths was melioidosis (n = 1,524 patients), followed by NTS infection (n = 1,005 deaths), *Vibrio* spp. infection (n = 172 deaths), *Streptococcus suis* infection (n = 85 deaths), *Corynebacterium diphtheriae* infection (n = 9 deaths), *Shigella* spp. infection (n = 7 deaths), *Salmonella enterica* serovar Paratyphi infection (n = 4 deaths), *S. enterica* serovar Typhi infection (n = 3 deaths), and *Neisseria meningitidis* infection (n = 3 deaths). None of 60 and 4 patients with culture-confirmed *N. gonorrhoeae* infection and *Brucella* spp. infection died in the hospital.

## Discussion

This study compared proportions, frequencies, all-cause in-hospital mortality and all-cause in-hospital mortality rate following AMR BSI between TCHs and SCHs in a LMIC. We show that the observed frequency of and all-cause in-hospital mortality rates following AMR BSI (per 100,000 admissions and per 100,000 patient-days) were considerably higher in TCHs than those in SCHs. The differences were also observed in the multivariable model adjusted for some confounders. The results support the needs to design antimicrobial stewardship (AMS) and infection prevention and control (IPC) in LMICs base on hospital size and type.

Our study highlights the capability of hospitals and national authorities in LMICs to esti-mate AMR frequencies as another crucial parameter to monitor the effectiveness of AMR interventions. For example, effectiveness of a nationwide intervention in Israel is shown by the reduction in frequency of nosocomial carbapenem-resistant *Enterobacteriaceae* infections from 55.5 to 11.7 cases per 100,000 patient-days [24]. An 80% reduction in MRSA BSI in England (defined as reported cases per 100,000 population per year and as reported cases per 100,000 bed-days) after a major public health infection prevention campaign also demon-strates the potential impact of coordinated interventions [25, 26].

The higher proportion (%) and frequencies of community-origin AMR BSI for CRAB and 3GCRKP in TCHs compared to those in SCHs could be due to higher proportion of health-care-associated infections in TCHs. It is likely that TCHs have a higher proportion of patients who are transferred from other hospitals (including from SCHs), who are receiving health care at end-stage renal facilities or long-term care facilities, and who are recently discharged from the hospitals. Routine hospital admission data used in our study could not identify those

conditions; therefore, those patients with BSI were categorized as community-origin BSIs in our reports. These findings suggest that TCHs may need to strengthen AMS and IPC, particularly on new inpatients who are at high risk of healthcare-associated infections.

No or minimal difference in proportion (%) of hospital-origin AMR BSI between TCHs and SCHs could be because the AMR prevalence in a country is likely driven by the contagion of AMR organisms within and between hospitals [27]. Although it is possible that higher use of antibiotics in TCHs may drive the emergence of AMR organisms (e.g. emergence of CREC and CRKP) and lead to the higher proportion of AMR BSI compared to those in SCHs, [28] that was not observed in our setting during the study period.

The strikingly higher frequencies of hospital-origin AMR BSI in TCHs than those in SCHs are likely caused by higher proportion of patients who had severe conditions or compromised immune systems, or required complex surgery, prolonged intubation or urinary catheters [11, 14]. The proportion of ICU beds in TCHs is also higher. Those conditions are likely driving both AMR and non-AMR hospital-acquired infections in our setting as shown by no or minimal difference in proportion of hospital-origin AMR BSI between TCHs and SCHs.

No or minimal difference in all-cause in-hospital mortality (%) following AMR BSI between TCHs and SCHs could be because care and antibiotics to be used against AMR BSI are not different between TCHs and SCHs in Thailand.

The higher all-cause in-hospital mortality rate following hospital-origin AMR BSI in TCHs than those in SCHs is, therefore, caused by higher frequencies of AMR BSI. These findings suggest that healthcare workers in TCHs will need to strengthen AMS and IPC, particularly among those are at high risk of hospital-acquired infections (HAI), more than those in SCHs. This is also because patients in TCHs are likely to be more complex than those in SCHs, and, as such, are at higher risk of HAI than those in SCHs.

For hospital-origin AMR BSI in SCHs, although CRAB, 3GCREC and 3GCRKP are the top three organisms with the highest frequencies, CRAB has the highest all-cause in-hospital mortality rate due to the higher all-cause in-hospital mortality (%) following CRAB BSI compared to those following 3GCREC BSI and 3GCRKP BSI.

Our findings of a high number of cases and all-cause in-hospital deaths following nontyphoidal *Salmonella* disease, [29, 30] melioidosis, [16, 31] *S. suis* infection, [32, 33] and *Vibrio* spp. Infection [34, 35] are consistent with previous research. Our study demonstrates that national statistics on multiple national notifiable bacterial diseases in LMICs could be improved by integrating information from readily available databases.

The strength of this study is that routine data used to compare AMR burdens between different hospital settings are from multiple sites in Thailand. Moreover, we have shown that the use of an automated surveillance system can readily generate useful statistics to understand AMR within a hospital and between hospitals in a country, and this empowers collaborative work and analyses across different settings both nationally and globally. The collaborative effort is essential to inform global burdens of AMR, which in turn are important statistics to support public health strategies to control spread of AMR and set priorities in resource allocation locally.

Our study has some limitations. Firstly, the findings from the study may not be generalisable to all LMICs. The differences in proportions, frequencies, all-cause in-hospital mortality and all-cause in-hospital mortality rates following AMR BSI observed in SCHs and TCHs could be confounded by many factors such as difference in patient characteristics, diagnostic stewardship (particularly blood culture utilization [22, 36]) and patient management (including AMS and IPC). For example, the differences in frequency and in-hospital mortality rate observed in our study could be confounded by the differences in the case mix between SCHs and TCHs. It is also possible that the AMS and IPC practices in Thailand are better in SCHs

compared to TCHs due to various reasons. These data were not available because our study was not designed to collected those data. The aim of this study was to examine the burden of AMR BSI observed in TCHs and compare that observed in SCHs using AMASS [15]. To understand the size of effect caused by the difference in hospital type alone, further studies using detailed clinical data, AMS and IPC data, and rigorous statistical approaches are needed. It is also possible that the AMS and IPC has already been tailored base on hospital size and type already. Regardless, our findings emphasize the need for further evaluation and improvement of all actions against AMR infections (including AMS and IPC) in TCHs, going beyond the current practices. Secondly, the sample size of SCHs and TCHs included in this study is small, and this may have limited the power to detect differences for organisms that are less predominate such as CRKP in the study setting. Thirdly, in this study we only included patients who were hospitalised and in-hospital mortality. Patients who had blood cultures taken either at community hospitals or the study hospitals but not hospitalised at the study hospitals were not included in the analysis. Fourthly, the mortality and mortality rate associated with AMR BSI reported are all-cause in-hospital mortality and mortality rate, and could be underestimated because some people, in the study area, preferred to die at home and were discharged against advice. The all-cause in-hospital mortality could also overestimate the impact of AMR infection. This is because a proportion of patients with AMR infection might die of other causes after the successful treatment of the AMR infection. This secondary data analysis was not designed to answer causal question but to demonstrate the potentials of using AMASS as a tool to empower routine AMR surveillance data understanding and sharing. Fifthly, the distinction between primary and secondary BSIs cannot be made without detailed, linked clinical and microbiology data. The BSIs in this study include both primary and secondary BSIs. Future surveillance systems should consider innovative methods that can integrate such information to ensure that these distinctions can be accurately made in LMICs. In addition, future studies with robust and comprehensive data collections that can capture all aspect of patient details, including the previous admission history and the previous culture results, the current diagnostic stewardship, AMS and IPC, and other confounding factors, are still needed.

In conclusion, we observed that the burden of AMR infections in TCHs is higher than that in SCHs in Thailand. This might also be occurring in other LMICs. Our results support similar evaluations in other LMICs and highlight the importance of tailoring infection control strategies based on hospital size and type, particularly when significant differences in the burden are observed. The frequency and in-hospital mortality rate of hospital-origin AMR BSI are important indicators and should be routinely measured to monitor the burden of AMR in every hospital with microbiology laboratories in LMICs.

## Supporting information

**S1 File.**
(DOC)

## Acknowledgments

We gratefully acknowledge the directors, epidemiological and laboratory team of general and regional hospitals for providing microbiological and hospital admission data and their administrative supports. The general and regional hospital network are comprised of Chorkaew Yangyuen (Samutprakarn Hospital), Aphinya Singkhongsin, Chanchira Chaichaem (Pranangklao Hospital), Phkaiwan Kropsanit (Pathumthani Hospital), Winai Suphapphot, Chaiwat Khaokaeo (Sena Hospital), Sasi Sichot (Phra Na Khon Sri Ayutthaya Hospital), Ratri Chalaemphak,

Benchawan Khaisongkhram (Angthong Hospital), Praphon Chinthanu, Phutthakhun Wong-suwan (Banmi Hospital), Pritsana wongnoi (Pranarai Maharaj Hospital), Witthaya Yotngoen (Singburi Hospital), Khongsak Sueachoi (Inburi Hospital), Pranom Chantharat (Jainad Nar-endra Hospital), Sangsan Sinbamrung, Duangkamon Chiratrachu (Phra Phutthabat Hospital), Waranya Sichanta (Saraburi Hospital), Panatda Thipruecha (Chonburi Hospital), Piyaphatcha Phongprasoet, Panatda Inphrom (Rayong Hospital), Pakkawi Siphueak, Ratchani Thamcham-rat (Prapokklao Hospital), Phuangphikun PhonPrasit (Trat Hospital), Bunga Chanlee (Bud-dhaSothorn Hospital), Wiphawadi Dongchan (Chao Phya Abhaibhubejhr Hospital), Haruethai Khunothai (Nakhonnayok Hospital), Atchara Ampere (Somdet Phrayupharacha SaKaeo), Saifon Sutchai, Prayut Kaeomalang, Nonglak Prayunsoet (Maharat Nakhon Ratcha-sima Hospital), Ratana Chiracharuporn (Buriram Hospital), Suriya Senthong (Surin Hospital), Sunthon Romniyaphet (Sisaket Hospital), Jintana Kanchanabat, Praweennuch Watanachai-prasert, Thanasith Sananmuang (Sunpasitthiprasong Hospital), Somphon Chankaeo (Yaso-thon Hospital), Wiraphon Khwamman (Chaiyaphum Hospital), Kraison Bunsam, Phonnatcha Katiwong (Amnat Charoen Hospital), Kanyaphak Phanchampa (Buengkan Hos-pital), Sutthiphong Phonbun (Nongbualamphu Hospital),Marisa Uton, Thitiphan Khunphu (Sirindhorn Hospital of Khon Kaen Province), Kriangkri Kongsuk, Ritthikorn DongLuang (Khon Kaen Hospital), Suthep Thipsawang, Kochnipa Kwawong, Nida Thanaphatphairot (UdonThani Hospital), Suphattra Likrachang, Kirana Phakdiburikun (Loei Hospital), Supha-kon Saenthamphon (Nongkhai Hospital), Suchitra Nasingkhan, Kanchanaphon Taratai (Mahasarakham Hospital), Witthaya Ratmaet (Roiet Hospital), Natthasorn Chawaninthawisut (Kalasin Hospital), Phuwanat PhothiChai (Sakonnakhon Hospital), Phinthip Saiklang (Nakhonphanom Hospital), Yutphon Mankhong, Yothin Tairayawong (Mukdahan Hospital), Warawan Inthip (Nakornping Hospital), Dr. Thiraphong Tatiyaphonkunthiraphong (Lam-phun Hospital), Thirin Ketwichit (Lampang Hospital), Yaowalak ChanDaeng (Uttaradit Hos-pital), Phana Thatsanawaythit, Itsareeya Boonrat (Phrae Hospital), Sopha Itsaranarongphan (Nan Hospital), Sanong Chaisue, Phanarat Phuangmali (ChiangKham Hospital), Chirawan Sithongphim (Phayao Hospital), Satorn Charatdamrongwat (Chiangrai Prachanukroh Hospi-tal), Jintana Phothip, Duangdi Chomphu (Srisangwan Hospital), Ladda Raden (Sawanpra-charak Hospital), Phitya Hema, Kanthika Ocharot, Mongkhon Uising (Uthaithani Hospital), Narong Mahayot (Kampangpetch Hospital), Onraphin Thiwai (Maesot Hospital), Preeyada Triprawat (Somdejprajaotaksin Maharaj Hospital), Kreangkrai Chatsut, Yuppharet Kaewpra-sern (Srisangworn Sukhothai Hospital), Meena Nakhon (Sukhothai Hospital), Aphinya Innoi, Thoranin Rakthanabodee (Buddhachinaraj hospital), Siwaphorn Phongchin (Phichit Hospi-tal), Jintana Phonphraram (Phetchabun Hospital), Somphon Niamlang (Damnoensaduak Hospital), Thanya Surakhamsang (Banpong Hospital), Thanyalak Borirak (Photharam Hospi-tal), Nopphon Siangchin (Ratchaburi Hospital), Ratchani Watthanayaem, Suwan Manutchan (Pahonpol Payuha Sena Hospital), Ekachai Photnanthawong (Makarak Hospital), Dr. Pornsak Thirathonbun (SomdejPrasangkharach17 Hospital), Narong Wongkanha (Chaophraya Yom-marat Hospital), Pitchayakhanid Yaemsoun, Pongphon Roeknaowarat (Nakhonpathom Hos-pital), Witthaya Sithong (Samutsakorn Hospital), Nonsi Sonthiyat (Kratumban Hospital), Narongchai SiamPhairi (Somdej Prabuddha Lertla Hospital), Chanthana Kalanuwat (Pra-jomklao Hospital), Saran Songsaeng (HuaHin Hospital), Chirawan Bunchusi (Maharaj Nakhonsithammarat Hospital), Chitchanin Niyomthai, Chutima Phayayam (Krabi Hospital), Boribun Chensamut (Takuapa Hospital), Aphisit Suwannarat, Suwanni Khwankhao, Phitya-porn Chunchu (Phang Nga Hospital), Phatcharin Yatraksa (Vachira Phuket Hospital), Jittima Thongnak (Koh Samui Hospital), Ratchanok Withunphan (Suratthani Hospital), Chuenkh-wan Kaeowichit, Phimnisa Phet (Ranong Hospital), Kritsanee Wichitakun (Chumporn Ketu-domsak Hospital), Sakda Khaophong (Songkla Hospital), Wichian Patangkaro, Nattawan

Chanmueang (Hat Yai Hospital), Nittaya Sakunsanti (Satun Hospital), Suriwan Phaksuphara, Umaporn Sina (Trang Hospital), Witthaya Wunchum (Phatthalung Hospital), Sirinthon Wongyoksuriya (Pattani Hospital), Mawin Deae, Ananni Sama (Betong Hospital), Wichai Wanmueang, Suphattra Mahachot (Yala Hospital), Arun Phutkaeo, Yukalipli Kaseng, Khodiyo Yamai (Narathiwat Ratchanakarin Hospital), Praphai Krirat, Naruemon Bunsiri (Sungai Kolok Hospital). We thank Saman Sayumphuruchinan (ED, MoPH), Wanwisa Khammak (Strategy and Planning Division, MoPH), Sittikorn Rongsumlee (MORU) and Prapass Wanapinij (MORU) for administrative and data management supports.

## Author Contributions

**Conceptualization:** Cherry Lim, Direk Limmathurotsakul.

**Formal analysis:** Cherry Lim, Direk Limmathurotsakul.

**Funding acquisition:** Direk Limmathurotsakul.

**Investigation:** Cherry Lim, Viriya Hantrakun, Preeyarach Klaytong, Chalida Rangsiwutisak, Ratanaporn Tangwangvivat, Chadaporn Phiancharoen, Pawinee Doung-ngern, Somkid Kripattanapong, Soawapak Hinjoy, Thitipong Yingyong, Archawin Rojanawiwat, Aekkawat Unahalekhaka, Watcharaporn Kamjumphol, Kulsumpun Khobanan, Pimrata Leethongdee, Narisorn Lorchirachoonkul, Suwimon Khusuwan, Suwatthiya Siriboon, Parinya Chamnan, Amornrat Vijitleela, Traithep Fongthong, Krittiya Noiprapai, Phairam Boonyarit, Voranadda Srisuphan, Benn Sartorius, John Stelling, Paul Turner, Nicholas P. J. Day, Direk Limmathurotsakul.

**Methodology:** Cherry Lim, Viriya Hantrakun, Preeyarach Klaytong, Chalida Rangsiwutisak, Ratanaporn Tangwangvivat, Chadaporn Phiancharoen, Pawinee Doung-ngern, Somkid Kripattanapong, Soawapak Hinjoy, Thitipong Yingyong, Archawin Rojanawiwat, Aekkawat Unahalekhaka, Watcharaporn Kamjumphol, Kulsumpun Khobanan, Pimrata Leethongdee, Narisorn Lorchirachoonkul, Suwimon Khusuwan, Suwatthiya Siriboon, Parinya Chamnan, Amornrat Vijitleela, Traithep Fongthong, Krittiya Noiprapai, Phairam Boonyarit, Voranadda Srisuphan, Benn Sartorius, John Stelling, Paul Turner, Nicholas P. J. Day, Direk Limmathurotsakul.

**Supervision:** Direk Limmathurotsakul.

**Writing – original draft:** Cherry Lim, Direk Limmathurotsakul.

**Writing – review & editing:** Cherry Lim, Viriya Hantrakun, Preeyarach Klaytong, Chalida Rangsiwutisak, Ratanaporn Tangwangvivat, Chadaporn Phiancharoen, Pawinee Doung-ngern, Somkid Kripattanapong, Soawapak Hinjoy, Thitipong Yingyong, Archawin Rojanawiwat, Aekkawat Unahalekhaka, Watcharaporn Kamjumphol, Kulsumpun Khobanan, Pimrata Leethongdee, Narisorn Lorchirachoonkul, Suwimon Khusuwan, Suwatthiya Siriboon, Parinya Chamnan, Amornrat Vijitleela, Traithep Fongthong, Krittiya Noiprapai, Phairam Boonyarit, Voranadda Srisuphan, Benn Sartorius, John Stelling, Paul Turner, Nicholas P. J. Day, Direk Limmathurotsakul.

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
