## [Decision Letter · Decision Letter 0]

1 Dec 2023

PONE-D-23-31373Frequency and mortality rate following antimicrobial-resistant bloodstream infections in tertiary-care hospitals compared with secondary-care hospitalsPLOS ONE

Dear Dr. Limmathurotsakul,

Thank you for submitting your manuscript to PLOS ONE. After careful consideration, we feel that it has merit but does not fully meet PLOS ONE’s publication criteria as it currently stands. Therefore, we invite you to submit a revised version of the manuscript that addresses the points raised during the review process.

We look forward to receiving your revised manuscript.

Kind regards,

Ali Amanati

Academic Editor

PLOS ONE

A clean copy of the edited manuscript (uploaded as the new *manuscript* file).

3. During our evaluation of the documents provided, we noted that the data were accessed for research purposes in November 2022, but this was after the project period had expired on the IRB documents supplied. Please could you provide the ethics approval extension documents for the study. If the document is in another language, please also provide an English translation. Please note that if these documents are not included when your manuscript is resubmitted, it may be rejected.

“The study was supported by the DDC, MoPH, Thailand, and Defense Threat Reduction Agency (DTRA), U.S.. This research was funded in part by the Wellcome Trust (224681/Z/21/Z and Wellcome Trust Institutional Translational Partnership Award-MORU). CL is supported by the Wellcome Trust (106680/B/14/Z). BS is supported by a grant from the UK Department of Health and Social Care using UK aid funding managed by the Fleming Fund (R52354 CN001). For the purpose of Open Access, the author has applied a CC BY public copyright licence to any Author Accepted Manuscript version arising from this submission.”

5. We noted in your submission details that a portion of your manuscript may have been presented or published elsewhere. [No. This work has not been previously published elsewhere. A pre-print version of this manuscript has been posted on medRxiv (https://www.medrxiv.org/content/10.1101/2023.02.07.23285611v2).] Please clarify whether this publication was peer-reviewed and formally published. If this work was previously peer-reviewed and published, in the cover letter please provide the reason that this work does not constitute dual publication and should be included in the current manuscript.

Additional Editor Comments:

Dear authors

Your manuscript [PONE-D-23-31373] has passed the review stage and is ready for ‎revision. ‎

Editorial comments

To ensure the Editor and Reviewers can recommend that your revised manuscript be ‎accepted, ‎‎please pay careful attention to each comment posted underneath ‎this email. This way we ‎can ‎avoid future clarifications and revisions, moving swiftly to ‎a decision.‎

‎1. Please provide a point-by-point response to the Editor and reviewer's comments

‎2. Please highlight all the amends on your manuscript with yellow color

‎3. Some grammatical and spacing errors need to be revised throughout the ‎manuscript

Editor comments:

There are significant objections to the statistical tests applied in this study.‎

As the second reviewer has pointed out, multivariable modeling ‎needs to be considered, and as the fourth reviewer has ‎pointed out, the significance of the obtained findings may be weakened ‎substantially without controlling potential confounders for investigating differences between hospital and also investigating factors associated with mortality. Please noticed that without considering advanced statistical revision I am unable to consider the manuscript further for peer review.‎

Truly yours

Reviewers' comments:

Reviewer's Responses to Questions

**Comments to the Author**

1. Is the manuscript technically sound, and do the data support the conclusions?

Reviewer #1: Yes

Reviewer #2: Partly

Reviewer #3: Partly

Reviewer #4: Partly

2. Has the statistical analysis been performed appropriately and rigorously? 

Reviewer #1: Yes

Reviewer #2: No

Reviewer #3: Yes

Reviewer #4: No

3. Have the authors made all data underlying the findings in their manuscript fully available?

Reviewer #1: Yes

Reviewer #2: Yes

Reviewer #3: Yes

Reviewer #4: Yes

4. Is the manuscript presented in an intelligible fashion and written in standard English?

Reviewer #1: Yes

Reviewer #2: Yes

Reviewer #3: Yes

Reviewer #4: Yes

5. Review Comments to the Author

Reviewer #1: Thanks to the research team for their valuable work. Please find my comments below:

1. In lines 220 to 227, you mentioned the hospitals included in the study, stating that out of 96 hospitals, 46 were excluded under certain conditions, and 49 studies were analyzed in your research, while it should have been 50. Please provide an explanation for this discrepancy regarding one study.

2. In the statistical section, it is necessary to clarify which variables have fixed effect(s) and which have random effect(s) in your mixed-effects logistic model.

3. Please be aware that p-values only indicate the level of significance, not the strength of the relationship. Therefore, consider this in all your tables and replace all values of <0.0001 with <0.001. NOTICE: "When p-values are displayed as very small numbers (e.g., 0.0001), it can be challenging for readers to quickly grasp their significance. It's more common to use the conventional format of displaying p-values as three decimal places (e.g., 0.001) for ease of interpretation."

4. The quality of your figures is significantly low, or they did not meet the desired quality in the submitted version, and I could not extract any useful information from them.. Please ensure that the figures meet the required quality standards or provide better versions in the submitted manuscript.

5. In the references section, you've used reports as sources. Please include relevant links for each reference to facilitate readers' access to the sources in your article.

Reviewer #2: I am impressed by the large amount of data collected during the study period 2012 to 2015; however, I recommend rejection due to the limited analysis performed and the similarities with previous studies.

Reviewer #3: Dear Authors

Kindly address the following few concerns

Line 154 correct the grammatical error " local importance" instead of "locally importance."

Line 215, second sentence is unclear, kindly revise.

LINE 398 "This is likely occurring in other LMICs” as part of the conclusion is too much of extrapolation from the current study under review. Rephrasing it to eg “This may be situation in other LMICs, or This might also be occurring in other LMICs” will be more scientifically acceptable.

Additional concerns

1. please include a section/paragraph that describes challenges or weakness or limitations of this study.

2. Are the patients' demographics accessible? We could make more meaningful inference from the comparative analysis with the demographic data. If this is not available, kindly include it in the weakness section.

Reviewer #4: Review of the manuscript frequency and mortality rate following antimicrobial-resistant blood stream infection in tertiary care hospital compared with the secondary care hospital.

This study addresses an important topic comparing the burden of antimicrobial resistant bloodstream infections between tertiary care and secondary care hospitals in Thailand. The study demonstrates the utility of using automated tools like AMASS for analyzing routine surveillance data to generate useful statistics on AMR infections. However, there are some limitations that should be acknowledged:

Major Comments:

1. More details needed on the data analysis plan - which specific metrics were compared between tertiary and secondary care hospitals and what statistical tests were used.

2. The results section overall is well-written, but could benefit from some sub-headings to improve readability by breaking it down into logical sections.

3. One limitation as acknowledged is that the findings may not be generalizable to other LMIC settings, so it may be useful to explicitly state this in the conclusion as well.

4. The paper does not provide information on the specific criteria used to classify community-origin and hospital-origin bloodstream infections (BSI), which may affect the accuracy of the results. Based only on admission date fails to account for healthcare exposures prior to admission. This could inflate community proportions at teaching hospitals receiving many transfers e.g. Patient with MRSA blood stream infection from previous hospitalization. More granular exposure data is needed

5. Designation of the Health care originated or associated blood stream infection relies on the fact that cultured organism should not be isolated from any site other than blood in order to be classified as BSI(blood stream infection) (www.cdc.gov/nhsn/pdfs/pscmanual/4psc_clabscurrent.pdf) during the infection window period. Absence of information about the culture results during the infection window period (with same organism) from sites other than blood can overestimate the proportion of blood stream infections

6. Major potential confounders like antibiotic usage, infection control practices are not adjusted for. This limits the ability to draw causal inferences about differences between hospital types.

7. Crude mortality rates do not adjust for important risk factors and likely oversimplify differences between complex tertiary vs general hospitals. More rigorous statistical approaches are needed to control for confounding

8. Relying on routinely collected data presents various quality issues like missing/inconsistent variables, coding errors, and variability in diagnostic/prescribing practices between sites. This reduces ability to accurately measure and compare burdens.

9. No validation is reported of automated classification (e.g. AMR vs susceptible). Misclassification bias could differentially affect estimates between settings

Minor Comments:

1. Standardize the formatting of abbreviations - write out full forms initially before introducing abbreviations.

2. Carefully proofread the manuscript to correct any minor grammar, spelling or punctuation errors.

3. Sample sizes: with only 15 TCHs and 34 SCHs, the study may be underpowered to detect some differences, especially for less common organisms. Larger multi-year datasets could provide more robust comparisons.

4. The study does not provide information on the specific interventions or strategies that could be implemented to address the differences in AMR burden between TCHs and SCHs.

5. The term AMR is not defined clearly whether it refers to an organism resistant to multiple families of antibiotics or just one

6. Mortality measures: all-cause in-hospital deaths only capture a fraction of outcomes and underestimate AMR impacts. Longer term and attributable mortality should also be examined.

7. Proportional measures like AMR proportions do not account for differences in total case volumes between hospitals. Higher proportions may still represent lower absolute burdens at smaller hospitals. Frequencies provide more clinically meaningful comparisons but are still prone to biases from differences in populations served.

8. Focus is on bacteremia but AMR impacts many infection types. Generalizing findings may overlook reservoir/transmission dynamics across all healthcare-associated infections.

Specific Comments:

1. Line 95-100 : The concept provided is misinterpreted , actually the reason for expressing the rates in 100s, 1000s or even 100000 is to provide the utility of comparison of rates across different locations and cohorts regardless of the size or services of the hospital https://www.cdc.gov/STD/Sassi/Module2/expression_of_rates.html

2. Line 181-192: it is not clear whether the BSI with common commensals are included in the analysis or not. according to CDC, 2 or more blood samples collected with common commensals and clinical picture constitutes BSI www.cdc.gov/nhsn/pdfs/pscmanual/4psc_clabscurrent.pdf

3. Line 194-200: the statistical analysis portion need elaboration as which of the strata of have less than 5 observations. There is mention of many statistical analysis. Author needs to clarify the data type used and rationale of using the statistical analyses.

4. Line 305-319: the text is out of the context of study as the study is about the frequency , proportion and burden of blood stream infections

5. Line 324-326: there is no assessment of AMS program or the infection control program for the participating facilities. So author cannot suggest if the practices are not tailored according to the scope of services already.

6. Line 338-343: this highlight the need of robust and comprehensive data collections that captures all aspects of patient detail including the previous admission history and the previous culture results

7. Line 344-349: no reference or bases provided for the assumptions

8. Line 354-256: contradicting statement to the earlier statement in line 324-326

9. Line 390-392: it could be overestimated as well. Since only the first isolate is taken into account there is no knowledge of how well the patient responding to the antibiotics and could die of cause other than BSI.

In summary, while a pragmatic first step, the methods have limitations for making robust policy inferences about AMR burdens in different hospital contexts. Greater standardization and more sophisticated statistical analyses are needed. Prospective collection of clinical and microbiological data would allow for more rigorous analyses of differences in AMR burdens and outcomes between hospital settings. Overall the manuscript makes a useful contribution on an important topic. Addressing the comments above would further improve it. I hope these suggestions are helpful for the authors. Please feel free to let me know if you need any clarification or have additional questions!

6. PLOS authors have the option to publish the peer review history of their article (what does this mean?). If published, this will include your full peer review and any attached files.

Reviewer #1: No

Reviewer #2: No

Reviewer #3: No

Reviewer #4: No

---

## [Author Response · Author response to Decision Letter 0]

15 Feb 2024

Response: Revised as suggested. 

A clean copy of the edited manuscript (uploaded as the new *manuscript* file).

Response: Revised as suggested. 

3. During our evaluation of the documents provided, we noted that the data were accessed for research purposes in November 2022, but this was after the project period had expired on the IRB documents supplied. Please could you provide the ethics approval extension documents for the study. If the document is in another language, please also provide an English translation. Please note that if these documents are not included when your manuscript is resubmitted, it may be rejected.

Response: The study (MUTM 2014-017-01) was approved on 4 March 2015, and was extended for 5 rounds until 3 March 2020. The documents confirming the extension of the study approval until 2020 in English have been attached. The study was closed as per protocol of the Ethics Committee of the Faculty of Tropical Medicine, Mahidol University, that the study could be closed when all data collection had been completed. Data analysis and manuscript submission could be performed after the study closure. A letter of confirmation of the discussion about data analysis regarding this study (MUTM 2014-017-01) from Ethics Committee of the Faculty of Tropical Medicine, Mahidol University has been attached. 

“The study was supported by the DDC, MoPH, Thailand, and Defense Threat Reduction Agency (DTRA), U.S.. This research was funded in part by the Wellcome Trust (224681/Z/21/Z and Wellcome Trust Institutional Translational Partnership Award-MORU). CL is supported by the Wellcome Trust (106680/B/14/Z). BS is supported by a grant from the UK Department of Health and Social Care using UK aid funding managed by the Fleming Fund (R52354 CN001). For the purpose of Open Access, the author has applied a CC BY public copyright licence to any Author Accepted Manuscript version arising from this submission.”

Response: We have added a statement in both the manuscript and the cover letter to acknowledge the role of the funders in the study.

5. We noted in your submission details that a portion of your manuscript may have been presented or published elsewhere. [No. This work has not been previously published elsewhere. A pre-print version of this manuscript has been posted on medRxiv (https://www.medrxiv.org/content/10.1101/2023.02.07.23285611v2).] Please clarify whether this publication was peer-reviewed and formally published. If this work was previously peer-reviewed and published, in the cover letter please provide the reason that this work does not constitute dual publication and should be included in the current manuscript.

Response: We confirm that the pre-print version posted on medRxiv was not peer-reviewed or formally published hence this work does not constitute dual publication. The medRxiv (https://www.medrxiv.org/content/about-medrxiv) is a free online archive for unpublished manuscripts (preprints) in medical, clinical and related health science. The current submitted manuscript has not been formally published in any platform. The below statement to clarify this is added in the cover letter: 

“A pre-print version that was not peer-reviewed or formally published of this manuscript has been posted on medRxiv (https://www.medrxiv.org/content/10.1101/2023.02.07.23285611v2)”

Response: DOIs to access our analysis data is now added in the main manuscript as below:

“The anonymous AMR surveillance reports generated from each hospital are open-access and available at https://figshare.com/s/c028f157c18a3cc06a82.”

Response: The captions for the supporting information files have been added at the end of the manuscript. 

Additional Editor Comments:

Dear authors

Your manuscript [PONE-D-23-31373] has passed the review stage and is ready for ‎revision. ‎

Editorial comments

To ensure the Editor and Reviewers can recommend that your revised manuscript be ‎accepted, ‎‎please pay careful attention to each comment posted underneath ‎this email. This way we ‎can ‎avoid future clarifications and revisions, moving swiftly to ‎a decision.‎

‎1. Please provide a point-by-point response to the Editor and reviewer's comments

‎2. Please highlight all the amends on your manuscript with yellow color

‎3. Some grammatical and spacing errors need to be revised throughout the ‎manuscript

Editor comments:

There are significant objections to the statistical tests applied in this study.‎

As the second reviewer has pointed out, multivariable modeling ‎needs to be considered, and as the fourth reviewer has ‎pointed out, the significance of the obtained findings may be weakened ‎substantially without controlling potential confounders for investigating differences between hospital and also investigating factors associated with mortality. Please noticed that without considering advanced statistical revision I am unable to consider the manuscript further for peer review.‎

Truly yours

Response: We appreciate invaluable suggestion and feedback from the editor and reviewers. We have now added multivariable mixed-effect regression models, as suggested. After adjusted for potential confounders, our findings still show differences in the frequency and mortality rates of AMR BSI between tertiary-care and secondary-care hospitals. This additional evidence supports our conclusion that targets of and resources for antimicrobial stewardship and infection control programs in LMICs might need to be tailored based on hospital type and size, as burden of AMR infections might differ by hospital setting. We have added the results from the multivariable models in both the main text of the manuscript and supporting documents. 

Reviewers' comments:

Reviewer's Responses to Questions

Comments to the Author

1. Is the manuscript technically sound, and do the data support the conclusions?

Reviewer #1: Yes

Reviewer #2: Partly

Reviewer #3: Partly

Reviewer #4: Partly

2. Has the statistical analysis been performed appropriately and rigorously? 

Reviewer #1: Yes

Reviewer #2: No

Reviewer #3: Yes

Reviewer #4: No

3. Have the authors made all data underlying the findings in their manuscript fully available?

Reviewer #1: Yes

Reviewer #2: Yes

Reviewer #3: Yes

Reviewer #4: Yes

4. Is the manuscript presented in an intelligible fashion and written in standard English?

Reviewer #1: Yes

Reviewer #2: Yes

Reviewer #3: Yes

Reviewer #4: Yes

5. Review Comments to the Author

Reviewer #1: Thanks to the research team for their valuable work. Please find my comments below:

1. In lines 220 to 227, you mentioned the hospitals included in the study, stating that out of 96 hospitals, 46 were excluded under certain conditions, and 49 studies were analyzed in your research, while it should have been 50. Please provide an explanation for this discrepancy regarding one study.

Response: We approached 96 hospitals and 95 of them agreed to participate in the study (Line 217). Of these 95 hospitals, 46 were excluded and 49 were analysed. We revised the sentence as below to clarify this:

“Ninety-five (99%) hospitals, out of 96 Thai hospitals that we approached, agreed to participate in the study.”

2. In the statistical section, it is necessary to clarify which variables have fixed effect(s) and which have random effect(s) in your mixed-effects logistic model.

Response: The lines below are now revised and added in manuscript to clarify mixed-effect models.

“We estimated the magnitude of differences in proportions and mortality of AMR BSI between SCHs and TCHs using mixed-effect logistic regression models for patients nested within hospital using the STATA command xtlogit. We estimated the magnitude of differences in frequency and mortality rate of AMR BSI between SCHs and TCHs using mixed-effect Poisson regression models for patients nested within hospital using the STATA command xtpoisson. We used mixed-effect models to estimate the fixed effects of hospital type while taking account for the random effects of patient within the same hospital. Multivariable mixed-effect models were also performed to control for other variables, including blood culture utilization rates of the hospitals and the health region where the hospitals located. These variables were included in the multivariable models because they could be associated with the hospital type and the outcome variables.”

3. Please be aware that p-values only indicate the level of significance, not the strength of the relationship. Therefore, consider this in all your tables and replace all values of <0.0001 with <0.001. NOTICE: "When p-values are displayed as very small numbers (e.g., 0.0001), it can be challenging for readers to quickly grasp their significance. It's more common to use the conventional format of displaying p-values as three decimal places (e.g., 0.001) for ease of interpretation."

Response: P-values in the text have been revised as suggested. 

4. The quality of your figures is significantly low, or they did not meet the desired quality in the submitted version, and I could not extract any useful information from them.. Please ensure that the figures meet the required quality standards or provide better versions in the submitted manuscript.

Response: Figures with improved resolutions have been uploaded. The figures with improved resolutions can be downloaded from the PLoS website using the links on the PDF. 

5. In the references section, you've used reports as sources. Please include relevant links for each reference to facilitate readers' access to the sources in your article.

Response: The links for references to reports have been added.

Reviewer #2: I am impressed by the large amount of data collected during the study period 2012 to 2015; however, I recommend rejection due to the limited analysis performed and the similarities with previous studies.

Response: Additional multivariable models are now added to control for variables that define the characteristics of the hospitals. The findings from the multivariable models have been added in both the main text of the manuscript and supporting documents. The results from the multivariable models still suggested differences in the observed frequency and mortality rates of AMR BSI between tertiary-care and secondary-care hospitals. The additional evidence supports our conclusion that the targets of and resources for antimicrobial stewardship and infection control programs in LMICs might need to be tailored based on hospital type and size, as burden of AMR infections might differ by hospital setting.

In addition, there are few studies comparing frequency and mortality rate following antimicrobial-resistant infections between tertiary-care hospitals and secondary-care hospitals in LMICs.

Reviewer #3: Dear Authors

Kindly address the following few concerns

Line 154 correct the grammatical error " local importance" instead of "locally importance."

Response: Revised as suggested.

Line 215, second sentence is unclear, kindly revise.

Response: Line 215 has been revised as below:

“Consent was not sought from the individual patients as this was a retrospective study. This approach was approved by the Ethical and Scientific Review committees.”

LINE 398 "This is likely occurring in other LMICs” as part of the conclusion is too much of extrapolation from the current study under review. Rephrasing it to eg “This may be situation in other LMICs, or This might also be occurring in other LMICs” will be more scientifically acceptable.

---

## [Decision Letter · Decision Letter 1]

13 Mar 2024

PONE-D-23-31373R1Frequency and mortality rate following antimicrobial-resistant bloodstream infections in tertiary-care hospitals compared with secondary-care hospitalsPLOS ONE

Dear Dr. Limmathurotsakul,

Thank you for submitting your manuscript to PLOS ONE. After careful consideration, we feel that it has merit but does not fully meet PLOS ONE’s publication criteria as it currently stands. Therefore, we invite you to submit a revised version of the manuscript that addresses the points raised during the review process.

We look forward to receiving your revised manuscript.

Kind regards,

*
**Ali Amanati**
*

**Academic Editor**

*
**PLOS ONE**
*

Journal Requirements:

**Additional Editor Comments:**

Dear authors

We are pleased to inform you that your manuscript has passed through ‎‎the ‎review stage and is ready for revision. The manuscript's overall ‎‎presentation ‎improved after amendments; ‎‎however, ‎still needs minor revision.‎

Editor comments:

Abstract:‎

‎Add corresponding statistics in the “results” section for the data presented in ‎the abstract section. Address the following shortcomings: ‎

a. “The frequencies of hospital-origin AMR BSI per 100,000 patient-days at risk ‎in TCHs were about twice that in SCHs for most pathogens under evaluation.”; ‎add frequencies and p. value. ‎

b. “All-cause in-hospital mortality (%) following hospital-origin AMR BSI was

not significantly different between TCHs and SCHs.”; add p. value.‎

c. “However, due to the higher frequencies, all-cause in-hospital mortality rates ‎following hospital-origin AMR BSI per 100,000 patient-days at risk were ‎considerably higher in TCHs for most pathogens.”; remove “However”; add ‎frequencies and p. value.‎

d. the following statement need corrections: ‎

‎“For example, the all-cause in-hospital mortality rate ...”; remove “For example, ”‎

e. Conclusion should be improved (both abstract and main text). It is a fact that AMR is varied in different ‎hospital settings. Based on the results of your study, you should highlight the ‎important indicators that found to be clinically more different and suggest ‎appropriate action to address issues.‎

Reviewers' comments:

Reviewer's Responses to Questions

**Comments to the Author**

1. If the authors have adequately addressed your comments raised in a previous round of review and you feel that this manuscript is now acceptable for publication, you may indicate that here to bypass the “Comments to the Author” section, enter your conflict of interest statement in the “Confidential to Editor” section, and submit your "Accept" recommendation.

Reviewer #1: (No Response)

Reviewer #3: All comments have been addressed

Reviewer #4: All comments have been addressed

2. Is the manuscript technically sound, and do the data support the conclusions?

Reviewer #1: Yes

Reviewer #3: Yes

Reviewer #4: Yes

3. Has the statistical analysis been performed appropriately and rigorously? 

Reviewer #1: No

Reviewer #3: Yes

Reviewer #4: Yes

4. Have the authors made all data underlying the findings in their manuscript fully available?

Reviewer #1: No

Reviewer #3: Yes

Reviewer #4: Yes

5. Is the manuscript presented in an intelligible fashion and written in standard English?

Reviewer #1: Yes

Reviewer #3: Yes

Reviewer #4: Yes

6. Review Comments to the Author

Reviewer #1: Your statistical section leaves me feeling disconnected for a couple of reasons. Firstly, there's no mention of demographic variables for the patients. Was this data unavailable or deemed unnecessary for your model? Secondly, the confounding variables you adjusted for aren't specified. Were these variables shared between patients and hospitals? Were they chosen based on their relevance to patients or hospitals? The only insights provided are tucked away in the supplementary file, in the footer of the tables.

Reviewer #3: No further comments at this stage. Authors have tried to address largely the concerns raised earlier.

Reviewer #4: All the previous comments have been addressed but the author have not mentioned which AMR proportions and Mortalities were analyzed using either the chi square test or fisher exact test or both.

line 222: author repeatedly mentioned "estimate of magnitude" but there is no mention of coefficient in result section , figures or in the supplemental material. kindly provide the output of analysis in stata showing coefficient.

7. PLOS authors have the option to publish the peer review history of their article (what does this mean?). If published, this will include your full peer review and any attached files.

Reviewer #1: No

Reviewer #3: No

Reviewer #4: No

---

## [Author Response · Author response to Decision Letter 1]

18 Mar 2024

Re: PONE-D-23-31373R1

Frequency and mortality rate following antimicrobial-resistant bloodstream infections in tertiary-care hospitals compared with secondary-care hospitals

PLOS ONE

Dear Editor and Reviewers,

We would like to thank the editor and all the reviewers for their time reading and commenting on our manuscript. We have revised the manuscript based on the feedback. In brief, the conclusion in the Abstract and Discussion sections are revised as suggested by the Editor. Also, frequencies, p-values, and mortality rates are included in the Abstract section as recommended by the Editor. The statistical analysis section is revised to improve clarity. 

We very much appreciate the suggestions and made changes in our manuscript as below:

Editor comments:

Abstract:‎

‎Add corresponding statistics in the “results” section for the data presented in ‎the abstract section. Address the following shortcomings: ‎

a. “The frequencies of hospital-origin AMR BSI per 100,000 patient-days at risk ‎in TCHs were about twice that in SCHs for most pathogens under evaluation.”; ‎add frequencies and p. value. ‎

Response: The frequencies and p values have been added as below:

“(for carbapenem-resistant Acinetobacter baumannii [CRAB]: 18.6 vs. 7.0, incidence rate ratio 2.77; 95%CI 1.72-4.43, p<0.001; for carbapenem-resistant Pseudomonas aeruginosa [CRPA]: 3.8 vs. 2.0, p=0.0073; third-generation cephalosporin resistant Escherichia coli [3GCREC]: 12.1 vs. 7.0, p<0.001; third-generation cephalosporin resistant Klebsiella pneumoniae [3GCRKP]: 12.2 vs. 5.4, p<0.001; carbapenem-resistant K. pneumoniae [CRKP]: 1.6 vs. 0.7, p=0.045; and methicillin-resistant Staphylococcus aureus [MRSA]: 5.1 vs. 2.5, p=0.0091).”

b. “All-cause in-hospital mortality (%) following hospital-origin AMR BSI was

not significantly different between TCHs and SCHs.”; add p. value.‎

Response: The p-values have been added. 

c. “However, due to the higher frequencies, all-cause in-hospital mortality rates ‎following hospital-origin AMR BSI per 100,000 patient-days at risk were ‎considerably higher in TCHs for most pathogens.”; remove “However”; add ‎frequencies and p. value.‎

Response: “However” and “For example” are now removed. The frequencies and p values are reported as below: 

“Due to the higher frequencies, all-cause in-hospital mortality rates following hospital-origin AMR BSI per 100,000 patient-days at risk were considerably higher in TCHs for most pathogens (for CRAB: 10.2 vs. 3.6, mortality rate ratio 2.77; 95%CI 1.71 to 4.48, p<0.001; CRPA: 1.6 vs. 0.8; p=0.020; 3GCREC: 4.0 vs. 2.4, p=0.009; 3GCRKP, 4.0 vs. 1.8, p<0.001; CRKP: 0.8 vs. 0.3, p=0.042; and MRSA: 2.3 vs. 1.1, p=0.023).”

d. the following statement need corrections: ‎

‎“For example, the all-cause in-hospital mortality rate ...”; remove “For example, ”‎

Response: “For example” has been removed.

e. Conclusion should be improved (both abstract and main text). It is a fact that AMR is varied in different ‎hospital settings. Based on the results of your study, you should highlight the ‎important indicators that found to be clinically more different and suggest ‎appropriate action to address issues.‎

Response: The conclusion has been improved as below in the Abstract and in the Discussion:

“The frequency and in-hospital mortality rate of hospital-origin AMR BSI are important indicators and should be routinely measured to monitor the burden of AMR in every hospital with microbiology laboratories in LMICs.”

Reviewers' comments:

Reviewer's Responses to Questions

Comments to the Author

1. If the authors have adequately addressed your comments raised in a previous round of review and you feel that this manuscript is now acceptable for publication, you may indicate that here to bypass the “Comments to the Author” section, enter your conflict of interest statement in the “Confidential to Editor” section, and submit your "Accept" recommendation.

Reviewer #1: (No Response)

Reviewer #3: All comments have been addressed

Reviewer #4: All comments have been addressed

2. Is the manuscript technically sound, and do the data support the conclusions?

Reviewer #1: Yes

Reviewer #3: Yes

Reviewer #4: Yes

3. Has the statistical analysis been performed appropriately and rigorously? 

Reviewer #1: No

Reviewer #3: Yes

Reviewer #4: Yes

4. Have the authors made all data underlying the findings in their manuscript fully available?

Reviewer #1: No

Reviewer #3: Yes

Reviewer #4: Yes

5. Is the manuscript presented in an intelligible fashion and written in standard English?

Reviewer #1: Yes

Reviewer #3: Yes

Reviewer #4: Yes

6. Review Comments to the Author

Reviewer #1: Your statistical section leaves me feeling disconnected for a couple of reasons. Firstly, there's no mention of demographic variables for the patients. Was this data unavailable or deemed unnecessary for your model? Secondly, the confounding variables you adjusted for aren't specified. Were these variables shared between patients and hospitals? Were they chosen based on their relevance to patients or hospitals? The only insights provided are tucked away in the supplementary file, in the footer of the tables.

Response: We would like to thank the reviewer for highlighting the potential confusing points in our manuscript. No individual patient-level data on demographic variables was collected. Our study is a secondary data analysis, which involved extracting aggregated statistics on AMR burden from routine AMR surveillance report generated by the 49 local hospitals. The sentence below is added to clarify:

“This secondary data analysis was not designed to answer causal question but to demonstrate the potentials of using AMASS as a tool to empower routine AMR surveillance data understanding and sharing.”

The sentence below is added in the “Statistical Analysis” section under the “Methods” to clarify the variables included in the multivariable analysis:

“Multivariable mixed-effect models were also performed to control for other variables, including blood culture utilization rates of the hospitals and the health region where the hospitals located.”

Reviewer #3: No further comments at this stage. Authors have tried to address largely the concerns raised earlier.

Reviewer #4: All the previous comments have been addressed but the author have not mentioned which AMR proportions and Mortalities were analyzed using either the chi square test or fisher exact test or both.

Response: We thank the reviewer for highlighting the confusing point on Chi square test and Fisher exact test. In the final analysis, neither Chi-square tests nor Fisher Exact tests were used. We have revised the sentence below to avoid confusion:

“We preliminarily compared AMR proportions and mortality using Chi-square or Fisher’s exact test when small samples (i.e. one or more expected values was <5 observations) and measurements from continuous variables using the Krustal-Wallis test. Then, we ..”

We also add all statistical tests used for all footnotes for clarity as below: 

Footnote of Figure 1: “Comparison was made using the Kruskal-Wallis test.”

Footnote of Figure 2: “Comparison was made using mixed-effect logistic or Poisson regression models of patients nested within hospitals.”

Footnote of Figure 3: “Comparison was made using mixed-effect logistic or Poisson regression models of patients nested within hospitals.”

All footnotes of supplementary tables. 

line 222: author repeatedly mentioned "estimate of magnitude" but there is no mention of coefficient in result section , figures or in the supplemental material. kindly provide the output of analysis in stata showing coefficient.

Response: The coefficients (from both univariable and multivariable models) are reported in the main text and supplementary materials in the form of incidence rate ratio (IRR) and mortality rate ratio (MRR), which are the exponentials of the coefficients. We decided not to report odds ratio of the logistic regression models to avoid confusion. We also removed the word “magnitude” to avoid confusion. 

7. PLOS authors have the option to publish the peer review history of their article (what does this mean?). If published, this will include your full peer review and any attached files.

Do you want your identity to be public for this peer review? For information about this choice, including consent withdrawal, please see our Privacy Policy.

Reviewer #1: No

Reviewer #3: No

Reviewer #4: No

All contributing authors have reviewed and concurred with the revised manuscript. 

Yours,

Direk Limmathurotsakul

---

## [Decision Letter · Decision Letter 2]

8 Apr 2024

PONE-D-23-31373R2Frequency and mortality rate following antimicrobial-resistant bloodstream infections in tertiary-care hospitals compared with secondary-care hospitalsPLOS ONE

Dear Dr. Limmathurotsakul,

Thank you for submitting your manuscript to PLOS ONE. After careful consideration, we feel that it has merit but does not fully meet PLOS ONE’s publication criteria as it currently stands. Therefore, we invite you to submit a revised version of the manuscript that addresses the points raised during the review process.

We look forward to receiving your revised manuscript.

Kind regards,

*
**Ali Amanati**
*

**Academic Editor**

*
**PLOS ONE**
*

Journal Requirements:

Additional Editor Comments:

Dear authors,

New comments were posted by the reviewer #1. So, the manuscripts require a round of revision.‎ Please provide a point-by-point response to the reviewer comments and highlight all the ‎amends on your manuscript with yellow color.‎

Yours

Reviewers' comments:

Reviewer's Responses to Questions

**Comments to the Author**

1. If the authors have adequately addressed your comments raised in a previous round of review and you feel that this manuscript is now acceptable for publication, you may indicate that here to bypass the “Comments to the Author” section, enter your conflict of interest statement in the “Confidential to Editor” section, and submit your "Accept" recommendation.

Reviewer #1: (No Response)

Reviewer #4: All comments have been addressed

2. Is the manuscript technically sound, and do the data support the conclusions?

Reviewer #1: Yes

Reviewer #4: Yes

3. Has the statistical analysis been performed appropriately and rigorously? 

Reviewer #1: No

Reviewer #4: Yes

4. Have the authors made all data underlying the findings in their manuscript fully available?

Reviewer #1: No

Reviewer #4: Yes

5. Is the manuscript presented in an intelligible fashion and written in standard English?

Reviewer #1: Yes

Reviewer #4: Yes

6. Review Comments to the Author

Reviewer #1: I appreciate the research team for answering the concerns. The authors' response does address some of the concerns by providing clarification on the nature of the data and the purpose of the study. However, there are still aspects that could be further explicit to improve the clarity and robustness of the study.

Demographic Variables: The authors clarify that individual patient-level data on demographic variables was not collected and that the study relied on aggregated statistics. While this explanation addresses the absence of demographic variables, it would be beneficial for the authors to explain why such data was not collected or why it was not deemed necessary for the model.

Confounding Variables: The authors mention that multivariable mixed-effect models were performed to control for other variables, including blood culture utilization rates of the hospitals and the health region where the hospitals were located. While this adds some clarity regarding the confounding variables, it would be helpful for the authors to elaborate on why these specific variables were chosen and how they were determined to be relevant to the analysis. Additionally, providing information on any other potential confounders that were considered but not included in the analysis would strengthen the robustness of the study.

Reviewer #4: no Comments . all the previous comments have been addressed .

7. PLOS authors have the option to publish the peer review history of their article (what does this mean?). If published, this will include your full peer review and any attached files.

Reviewer #1: No

Reviewer #4: No

---

## [Author Response · Author response to Decision Letter 2]

14 Apr 2024

Re: PONE-D-23-31373R2

Frequency and mortality rate following antimicrobial-resistant bloodstream infections in tertiary-care hospitals compared with secondary-care hospitals

PLOS ONE

Dear Editor and Reviewers,

We would like to thank the editor and all the reviewers for their time reading and commenting on our manuscript. We have revised the manuscript based on the feedback. 

We very much appreciate the suggestions and made changes in our manuscript as below:

Reviewer #1: I appreciate the research team for answering the concerns. The authors' response does address some of the concerns by providing clarification on the nature of the data and the purpose of the study. However, there are still aspects that could be further explicit to improve the clarity and robustness of the study.

Demographic Variables: The authors clarify that individual patient-level data on demographic variables was not collected and that the study relied on aggregated statistics. While this explanation addresses the absence of demographic variables, it would be beneficial for the authors to explain why such data was not collected or why it was not deemed necessary for the model.

Response: We are grateful for the advice. We have revised and elaborated the details of data collected from the study hospitals and used for this study in the method section as follow, “Variables in the microbiology data file included patient hospital number (HN), specimen type, specimen collection date, culture result, and antibiotic susceptibility testing result, and each row contained information for each specimen. Variables in the hospital admission data file included HN, admission date, discharge date, and in-hospital discharge outcome, and each row contained information for each admission.” 

We have also revised and elaborated this limitation in the discussion section as follow, “These data were not available because our study was not designed to collected those data. The aim of this study was to examine the burden of AMR BSI observed in TCHs and compare that observed in SCHs using AMASS.15 To understand the size of effect caused by the difference in hospital type alone, further studies using detailed clinical data, AMS and IPC data, and rigorous statistical approaches are needed.”

Confounding Variables: The authors mention that multivariable mixed-effect models were performed to control for other variables, including blood culture utilization rates of the hospitals and the health region where the hospitals were located. While this adds some clarity regarding the confounding variables, it would be helpful for the authors to elaborate on why these specific variables were chosen and how they were determined to be relevant to the analysis. Additionally, providing information on any other potential confounders that were considered but not included in the analysis would strengthen the robustness of the study.

Response: We are grateful for the advice. We have revised and elaborated the details of data collected from the study hospitals in details in the method section as follow, “We adjusted for blood culture utilization rate because if blood culture utilization rate was low, the observed AMR proportions could be higher than the true susceptibility profiles of pathogenic organisms.22 Additionally, the observed AMR frequency per 100,000 admissions and per 100,000 patient-days could be lower than the true incidence rate of AMR infections.22 We adjusted for health regions to control for potential variations in the AMR proportions and frequency across different regions. These regional differences could be influenced by other factors such as habits of antibiotic use and economic levels which were not explored in this study. There were no other potential confounders that were evaluated in the analysis.”

All contributing authors have reviewed and concurred with the revised manuscript. 

Yours,

Direk Limmathurotsakul

---

## [Decision Letter · Decision Letter 3]

22 Apr 2024

Frequency and mortality rate following antimicrobial-resistant bloodstream infections in tertiary-care hospitals compared with secondary-care hospitals

PONE-D-23-31373R3

Dear Dr. Direk Limmathurotsakul,

We’re pleased to inform you that your manuscript has been judged scientifically suitable for publication and will be formally accepted for publication once it meets all outstanding technical requirements.

Kind regards,

*
**Ali Amanati**
*

**Academic Editor**

*
**PLOS ONE**
*

Additional Editor Comments (optional):

I read the revised manuscript ‎

I have no further comments to add. I thank the authors for their detailed ‎‎replies to the reviewers' comments.‎

Reviewers' comments:

Reviewer's Responses to Questions

**Comments to the Author**

1. If the authors have adequately addressed your comments raised in a previous round of review and you feel that this manuscript is now acceptable for publication, you may indicate that here to bypass the “Comments to the Author” section, enter your conflict of interest statement in the “Confidential to Editor” section, and submit your "Accept" recommendation.

Reviewer #1: All comments have been addressed

2. Is the manuscript technically sound, and do the data support the conclusions?

Reviewer #1: Yes

3. Has the statistical analysis been performed appropriately and rigorously? 

Reviewer #1: Yes

4. Have the authors made all data underlying the findings in their manuscript fully available?

Reviewer #1: Yes

5. Is the manuscript presented in an intelligible fashion and written in standard English?

Reviewer #1: Yes

6. Review Comments to the Author

Reviewer #1: (No Response)

7. PLOS authors have the option to publish the peer review history of their article (what does this mean?). If published, this will include your full peer review and any attached files.

Reviewer #1: No

---

## [Editor Report · Acceptance letter]

29 Apr 2024

PONE-D-23-31373R3 

PLOS ONE

Dear Dr. Limmathurotsakul, 

I'm pleased to inform you that your manuscript has been deemed suitable for publication in PLOS ONE. Congratulations! Your manuscript is now being handed over to our production team.

Kind regards, 

on behalf of

Professor Ali Amanati 

Academic Editor

PLOS ONE